# Association of multidrug-resistant bacteria and clinical outcomes in patients with infected diabetic foot in a Peruvian hospital: A retrospective cohort analysis

Marlon Yovera-Aldana[1]*, Paola Sifuentes-Hermenegildo[2,3], Martha Sofia Cervera-Ocaña[4], Edward Mezones-Holguin[5,6]

1 Grupo de Investigación de Neurociencias, Efectividad Clínica y Salud Pública, Universidad Científica del Sur, Lima, Peru, 2 Departamento de Medicina, Servicio de Endocrinología, Hospital María Auxiliadora, Lima, Peru, 3 Red de Eficacia Clínica y Sanitaria, Lima, Perú, 4 Facultad de Ciencias Médicas, Universidad César Vallejo, Trujillo, Peru, 5 Centro de Estudios Económicos y Sociales en Salud, Universidad San Ignacio de Loyola, Lima, Peru, 6 Soluciones de Epi-gnosis, Piura, Peru

* myovera@cientifica.edu.pe

## Abstract

### Objective

To evaluate the association of multidrug-resistant bacteria (MDRB) and adverse clinical outcomes in patients with diabetic foot infection (DFI) in a Peruvian hospital.

### Materials and methods

This retrospective cohort study evaluated patients treated in the Diabetic Foot Unit of a General Hospital in Lima, Peru. MDRB was defined by resistance to more than two pharmacological groups across six clinically significant genera. The primary outcome was death due to DFI complications and/or major amputation. Other outcomes included minor amputation, hospitalization, and a hospital stay longer than 14 days. Relative risks were estimated using Poisson regression for all outcomes.

### Results

The study included 192 DFI patients with a mean age of 59.9 years; 74% were males. A total of 80.8% exhibited MDRB. The primary outcome had an incidence rate of 23.2% and 5.4% in patients with and without MDRB, respectively (p = 0.01). After adjusting for sex, age, bone involvement, severe infection, ischemia, diabetes duration, and glycosylated hemoglobin, MDRB showed no association with the primary outcome (RR 3.29; 95% CI, 0.77–13.9), but did with hospitalization longer than 14 days (RR 1.43; 95% CI, 1.04–1.98).

### Conclusions

Our study found no association between MDRB and increased mortality and/or major amputation due to DFI complications, but did find a correlation with prolonged hospitalization. The high proportion of MDRB could limit the demonstration of the relationship. It is urgent to

**Data Availability Statement:** All relevant data are within the manuscript and its Supporting Information files.

**Funding:** The author(s) received no specific funding for this work.

**Competing interests:** The authors have declared that no competing interests exist.

apply continuous evaluation of bacterial resistance, implement a rational plan for antibiotic use, and maintain biosafety to confront this threat.

## Introduction

Diabetic Foot infection (DFI) is one of the leading causes of hospitalization in diabetic foot care units, with 15–20% of cases requiring major amputation [1,2]. In Peru, it accounts for 18.9% of diabetes mellitus-related hospital admissions, and 61% of these cases develop sepsis [3]. Treatment costs are five times higher for patients who develop this complication, which impacts the healthcare system and patients' quality of life, particularly in developing countries [4]. Given the magnitude and significant impact of this complication, effective and personalized treatment is needed.

Antibiotic resistance poses a threat to the successful treatment of DFI [5]. The prevalence of Multidrug-Resistant Bacteria (MDRB) varies from 14% to 66% [6,7], depending on the country studied. MDRB is associated with protracted recovery, increased need for surgical procedures, prolonged hospitalization, and higher treatment costs [8]. Additionally, it results in increased protein intake requirements, higher oxygen consumption, disturbed glycemic control, and reduced hemoglobin levels [9]. As these are polymicrobial infections, the use of broad-spectrum antibiotics is recommended for severe infections, subsequently deescalating according to the antibiogram [10]. For a proper DFI approach, it is mandatory to determine the antimicrobial resistance profile at each healthcare facility in order to devise a treatment algorithm based on those results [11].

The influence of MDRB on the clinical outcomes of DFI is not well understood [7], and there is limited research conducted on this subject in Latin America [3]. Factors such as inappropriate antibiotic selection due to the lack of an antibiogram, reduced antibiotic concentration in tissues caused by peripheral ischemia, and colonization by resistant strains due to repeated hospitalizations or prolonged treatments have been documented [12]. Furthermore, at the local level, only 30% of outpatient individuals achieve an HbA1c level below 7%, and a mere 10% attain comprehensive metabolic control (involving lipids, weight, blood pressure, and blood glucose). Consequently, adverse clinical outcomes can potentially be expected when faced with an MDRB infection [13]. This study aims to assess the relationship between MDRB and its outcomes in DFI patients at a public hospital in Peru.

## Material and methods

### Study design and setting

We conducted a retrospective cohort study on patients with DFI. We used secondary data from patients who attended the Diabetic Foot Unit (DFU) at the Maria Auxiliadora Hospital in Lima, Peru, between January 2017 and December 2019. This hospital is in southern Lima and provides medical care to 2.5 million economically disadvantaged people. The DFU of the Endocrinology service has been the first line of treatment for DFI patients since 2015. Based on the severity, the DFU determines whether hospitalization or outpatient management is required.

## Patients

We included patients diagnosed with DFI as per the Infectious Diseases Society of America (IDSA) classification [10]. Additionally, those with a positive culture taken by the DFU staff within the first 48 hours of contact were included. We excluded patients with the following criteria: extensive necrosis caused by severe ischemia, inability to obtain culture samples, culture samples obtained via swabs or aspiration, follow-up period of less than one month if not hospitalized, infected peripheral venous insufficiency, and pressure ulcers.

## Sample size

After applying the inclusion and exclusion criteria, we obtained a sample size of 192 patients. The sample's statistical power was 88%, calculated based on an 82% proportion of the primary outcome among subjects with MDR and 78% in subjects without MDR, at a confidence level of 95%, with a 4:1 ratio between MDR and non-MDR subjects.

## Variables

**Multidrug-Resistant Bacteria (MDRB).** MDRB was defined as the lack of susceptibility to at least one agent in three or more classes of antimicrobials for each bacterial genus: *Staphylococcus sp.*, *Enterococcus sp.*, *Enterobacteriaceae*, *Pseudomonas sp.*, and *Acinetobacter sp.*. Innate resistances to some drugs were not considered for this definition. **S1–S5 Tables** show the criteria used to determine resistance for each bacterial genus [14].

**Clinical outcomes.** The primary outcome was a composite of major leg amputation or death during hospitalization or within the first 30 days after discharge. Any surgery above the ankle was considered a major amputation. In **S6 Table**, we present the codes for procedures classified as major amputation according to the Current Procedural Terminology (CPT). As a secondary outcome, we assessed the individual components of the primary outcome. We also considered minor leg amputations, which included phalangeal disarticulation, beam amputation, transmetatarsal amputation, Lisfranc, and Syme amputation. Another secondary outcome was the indication for inpatient care for intravenous antibiotic treatment and prolonged hospitalization (hospital stays longer than 14 days).

**Other variables.** We described the demographics and clinical characteristics of lesions and patients upon admission to the DFU. In the demographic variables, we described gender (male/female), age ($\geq$ 60/$<$ 60 years), and duration of diabetes ($\geq$ 10 years/ $<$ 10 years). Regarding foot characteristics, we described the Wagner classification (stage 1, 2, 3, 4), infection severity based on the Infectious Disease Of Society American (IDSA) Guideline of DFI (mild, moderate, severe) [10]; peripheral arterial disease (PAD) determined from arterial plethysmography reports (positive result if previous history or biphasic, monophasic, or absence of wave pattern/ negative if triphasic pattern), type of ulcer (new/ re-ulceration); ulcer area calculated using the ellipse formula ($\geq$ 10 cm2 /$<$ 10 cm2), peripheral neuropathy (Yes if previous history or recorded as loss of protective sensation evaluated with a monofilament). Additionally, we described the San Elian score, which includes description of depth, infection, ischemia, neuropathy, location, topography, number of zones, edema, and degree of injury inflammation (moderate or $<$ 20 / severe or $\geq$20) [15] For informational purposes, we describe the components and categories of the scales used and others that were not available in **S7 Table**.

We recorded the following clinical history details: previous major amputation or surgery above the ankle (yes/no), any history of minor amputation or surgery below the ankle due to diabetic foot issues (yes /no), any previous hospitalization for diabetic foot-related reasons (yes/no); previous antibiotic therapy within the last year (yes/no), and previous DFI within the last year (yes/no). Also, we recorded previous diabetic retinopathy (yes/no), previous chronic

kidney disease by CKD-EPI formula ($\geq$60 ml/min/24 h /$<$ 60 ml/min/24 h) [16], hospitalization duration and total follow-up period in days.

In the laboratory results, we described hemoglobin ($< 10/ \geq 10$ g/dl), albumin ($< 3.5/ \geq$ 3.5 g/dl), and glycosylated hemoglobin ($< 7/ \geq 7$%). HbA1c values up to one month old were accepted. Additionally, we recorded the number of bacterial strains (monomicrobial/ polymicrobial).

## Procedures

Authorization was obtained from the Endocrinology department head to access DFU data. The endocrinology service continuously records patient outcomes for administrative purposes. The spreadsheet provided did not contain personally identifiable information. A first data request was made in February 2019 and then in December 2019. Two authors completed other data of interest by reviewing the medical records from February 2019 to January 2020 as needed (S8 Table).

**Management guide.** The endocrinology service staff applied diagnostic and therapeutic procedures following the International Working Group of Diabetic Foot Guide for the management of infection, peripheral arterial disease, and ulcers [17]. The team comprised four doctors and two nurses trained in comprehensive care. At the María Auxiliadora hospital, patients with acute infection, whether admitted through the clinic or the emergency department, are preferentially managed by a member of the DFU team. The decision for hospitalization or outpatient care depends on the severity of the infection, the patient's age, and environmental factors such as family support.

Upon admission, all patients undergo a wound evaluation according to standardized formats. An arterial Doppler ultrasound of the affected limb is performed to assess arterial waveform characteristics related to the injury. As per protocol, the infected lesion is debrided within the first 48 hours after admission; However, swab cultures are not performed. Only transportation and cultivation methods for aerobic strains are available. The culture is taken by biopsy of the debrided wound bed. Transportation is carried out in sterile warehouses. The antibiotic is started according to the IDSA scale for empirical treatment. Antibiotics are adjusted based on culture results obtained after 72 hours. During hospitalization, the patient receives wound care three to four times weekly from the DFU team, with bedside surgical debridement and advanced wound healing using silver hydrogels or other similar modalities.

**Microbiological analysis.** Culture results were obtained from clinical records, encompassing all identified strains. The endocrinology service staff performed culture samples via tissue biopsy following wound debridement using the standard procedures [18]. Subsequently, the hospital laboratory staff conducted the bacterial identification analysis and antibiotic susceptibility tests using the automated VITEK® 2 system (BioMérieux Laboratory, Argentina) [19].

## Statistical analysis

Clinical and laboratory characteristics were quantitatively described using either mean with standard deviation or median with interquartile range, depending on the normality assessed by the Shapiro-Wilk test. Categorical variables were presented with absolute and relative frequencies. Moreover, isolated bacteria were characterized by percentages according to the Gram stain type and genus. To assess differences between groups with and without MDRB, we used the Student's T-test or Mann-Whitney U test for numerical variables depending on distribution. For categorical variables, we employed the Pearson's Chi-square test or Fisher's exact test. We also assessed the incidence of the primary composite outcome based on demographic and clinical characteristics.

Multivariate analysis was performed using a generalized linear model with a Poisson function, logarithmic link, and robust variance to determine the relative risk (RR) for the primary and secondary outcomes. An unadjusted model and three adjusted models were developed based on epidemiological variables. Model 1 was adjusted for age and sex, Model 2 for bone involvement, severe infection, and peripheral arterial disease, and Model 3 for diabetes duration and HbA1C levels. The analysis was conducted using the STATA® software (version 15.1, Texas, USA), with a significance level of 5% applied for all hypothesis tests.

**Ethics.** The Institutional Ethics Committee for Research at María Auxiliadora Hospital approved the development of this research project under the code HA/CIEI/019/19. The data provided by the Endocrinology Department did not contain any personally identifiable information.

## Results

The initial database comprised 582 subjects evaluated by the DFU between 2017 and 2019. Based on eligibility criteria, we excluded 390 patients, primarily due to the absence of a bacterial culture. Ultimately, 192 patients were included in the study (**Fig 1**).

Out of the total, 155 patients (80.8%) had a MDRB infection, 74% of whom were males, with an average age of 59.9 ± 12.9 years. The median duration of diabetes mellitus was 12 years, and the average HbA1c was 10%, with 81% of patients having an HbA1c level of ≥7. Notably, 28.1% of the patients developed a recurrent ulcer. According to the Wagner classification of DFI, 50.5% were classified as grade IV, and 88% had a moderate to severe infection according to the IDSA guidelines. Additional demographic and clinical characteristics of the patients are shown in **Table 1**.

Of a total of 236 isolated bacteria, 82% were MDR. The most common MDR bacteria were *Enterobacteriaceae* (51.3%), *Staphylococcus aureus* (28.4%), and *Enterococcus sp.* (8.9%). Gram-negative bacteria accounted for 61% of the cases, and only 37 (19.3%) had a polymicrobial culture. A detailed description of the isolated MDR organisms is provided in **Table 2**.

The primary outcome (death and/or major amputation) occurred in 18.9% of all patients. The mortality rate was 2.1%, and major amputation was observed in 17.7%. Regarding secondary outcomes, 67% of the general population required hospitalization. Among those hospitalized, 80% had a hospital stay longer than 14 days (**Table 3** **and S1 Fig**).

The demographic characteristics associated with the primary composite outcome were age over 60 years, duration of diabetes over 10 years, ulcer area larger than 10 cm$^2$, depth of the lesion according to Wagner classification [20], infection grade according to IDSA [10], presence of peripheral arterial disease, San Elian score ≥ 20 [15], hemoglobin <10 g/dl, albumin <3.5 g/dl, HbA1c ≥7%, and polymicrobial culture (**Table 4**).

In the crude model, for patients with MDRB, the incidence of the primary composite clinical outcome increased by 3.3 times. However, when adjusted for epidemiological variables, no association was evidenced in any of the three proposed models. Among the secondary outcomes, only a hospital stay longer than 14 days was associated with MDRB in the adjusted models. The outcomes and their adjusted RR are presented in **Table 5**.

## Discussion

### Main results

In this study, we found no association between MDRB and the primary outcome of death related to diabetic foot and/or major amputation, but we did find an association with a hospital stay longer than 14 days.

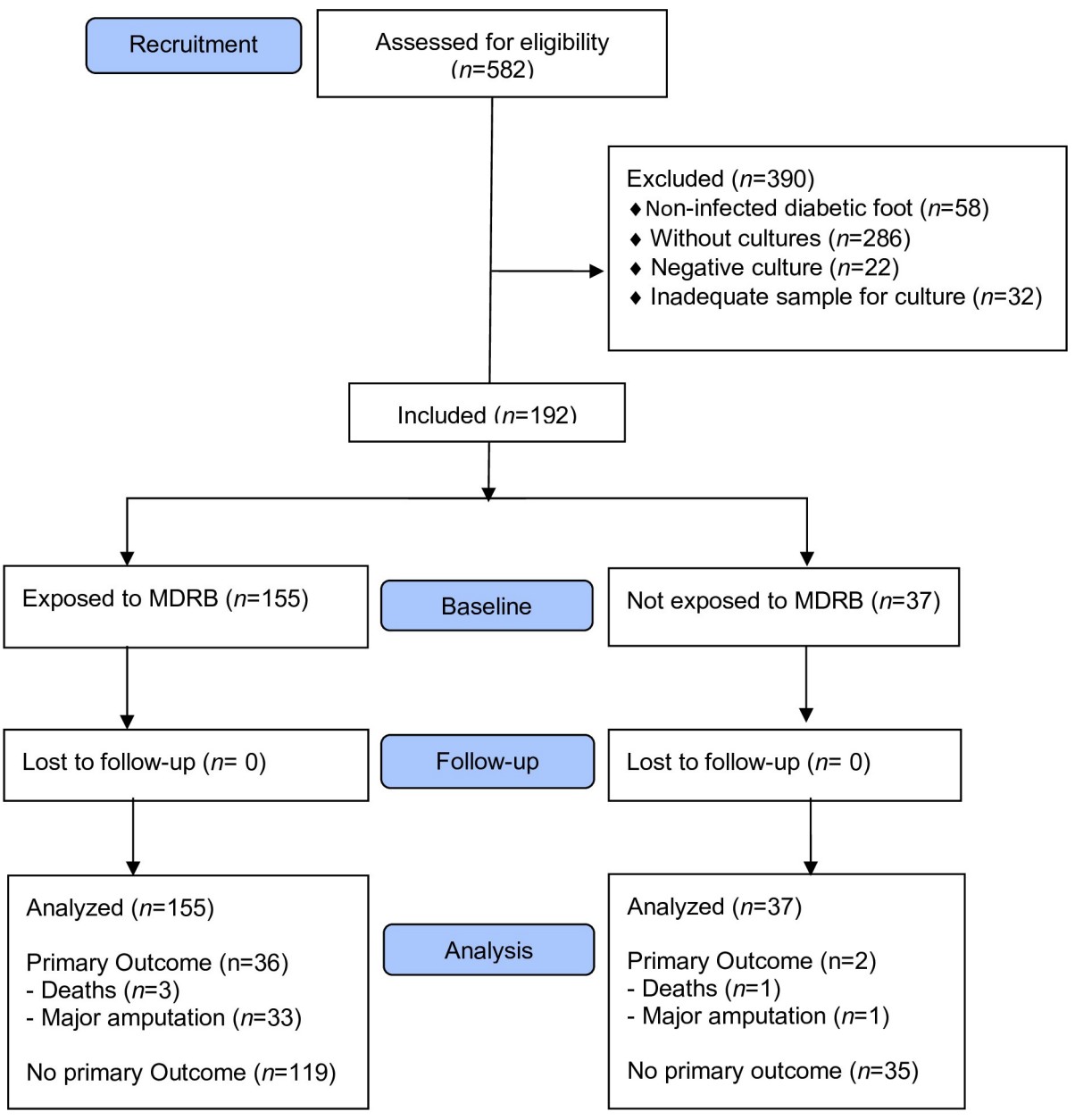

**Fig 1. Flowchart of patients included in the study.**

## Comparison with other studies

Different studies have shown controversial results regarding the influence of MDRB on the clinical course of DFI. Gupta et al. [7] and Richard et al. [21] concluded that MDRB is not associated with clinical outcomes in patients with DFI. In these studies, the lack of association was attributed to early aggressive treatment, antibiotic therapy adjusted to microbiological findings, and the small number of patients with MDRB included. In our study, after multivariate analysis, no association was found between MDRB and a higher risk of major amputation. However, Matta et al.'s systematic review did find a higher prevalence of amputation in DFI with MDRB. However, these studies only performed univariate analysis and considered that

**Table 1. Clinical and demographic characteristics according to MDRB status.**

| Variable | Total n(%) | MDRB (+) n(%) | MDRB (-) n(%) | p-value |
|---|---|---|---|---|
| **General** | 192 (100) | 155 (80.8) | 37 (19.3) | |
| **Demographic** | | | | |
| Male | 142 (74.0) | 112 (72.3) | 30 (81.1) | 0.270 |
| Age (years), mean ± SD | 59.9 ± 12.9 | 60.8 ± 12.9 | 56.5 ± 12.8 | 0.070 |
| Age ≥ 60 years | 96 (50.0) | 83 (53.6) | 13 (35.1) | **0.040** |
| DM duration (years), Med (IQR) | 12 (8 to 20) | 12 (9 to 20) | 12 (8 to 17) | 0.720 |
| **Diabetic foot ulcer** | | | | |
| Re-ulceration | 54 (28.1) | 44 (28.4) | 10 (27.0) | 0.870 |
| Total ulcer area (cm$^2$), Med (IQR) | 15 (6 to 30) | 15 (6 to 32) | 18 (6 to 24) | 0.830 |
| Wagner classification | | | | |
| 1 | 5 (2.6) | 4 (2.6) | 1 (2.7) | 0.150 |
| 2 | 18 (9.4) | 12 (7.7) | 6 (16.2) | |
| 3 | 72 (37.5) | 55 (35.5) | 17 (46.0) | |
| 4 | 97 (50.5) | 84 (54.2) | 13 (35.1) | |
| IDSA infection grade | | | | |
| Mild | 23 (12.0) | 15 (9.7) | 8 (21.6) | 0.080 |
| Moderate | 122 (63.5) | 104 (67.1) | 18 (48.6) | |
| Severe | 47 (24.5) | 36 (23.2) | 11 (29.7) | |
| Peripheral arterial disease | 71 (37.0) | 62 (40.0) | 9 (24.3) | 0.070 |
| Diabetic neuropathy | 186 (96.9) | 151 (97.4) | 35 (94.6) | 0.370 |
| San Elian score, mean ± SD | 18.5 ± 3.0 | 18.6±3.0 | 18.4±3.4 | 0.680 |
| San Elian score ≥ 20 | 70 (36.5) | 58 (37.4) | 12 (32.4) | 0.570 |
| **Comorbidities** | | | | |
| Previous major amputation due to diabetic foot | 14 (7.3) | 12 (7.7) | 2 (5.4) | 0.870 |
| Any previous minor amputation due to diabetic foot | 66 (34.6) | 55 (35.7) | 11 (29.7) | 0.490 |
| Any previous diabetic foot hospitalization | 36 (18.8) | 32 (20.7) | 4 (10.8) | 0.170 |
| Antibiotic therapy within the last year | 68 (35.4) | 52 (33.6) | 16 (43.2) | 0.270 |
| Diabetic foot infection within the last year | 62 (32.3) | 53 (34.2) | 9 (24.3) | 0.250 |
| GFR, mean ± SD | 88.7 ± 32 | 87.2 ± 32.0 | 95.1 ± 29.8 | 0.180 |
| **Laboratory Findings** | | | | |
| Hemoglobin (g/dL), mean ± SD | 10.7 ± 1.9 | 10.6 ± 2.0 | 11.0 ± 1.8 | 0.110 |
| Leukocytes (cells x10$^3$/mm$^3$), Med (IQR) | 11.9 (8.4 to 16.6) | 11.9 (8.4 to 16.6) | 11.5 (8.4 to 16.9) | 0.980 |
| Albumin (g/dL), mean ± SD | 3.0 ± 0.6 | 3.0 ± 0.6 | 3.2 ± 0.6 | 0.140 |
| HbA1c (%), mean ± SD | 10.1 ± 3.0 | 9.7 ± 2.9 | 11.7 ± 2.9 | **0.006** |
| HbA1c ≥ 7% | 128 (81.0) | 97 (77.6) | 31 (93.9) | **0.030** |
| **Microbiological** | | | | |
| Polymicrobial | 37 (19.3) | 33 (21.3) | 4 (10.8) | 0.140 |
| **Time** | | | | |
| **Total follow-up (days)** | 163 [56.5 to 439] | 172 [54 to 448] | 137 [89 to 397] | |
| **Length of hospital stay (days)** | 22 [15 to 32] | 23.5 [15 to 32] | 17.5 [12 to 31] | |

Data are presented as n (%). DM: Diabetes Mellitus. SD: Standard Deviation. Med: Median. IQR: Interquartile Range. GFR: Glomerular Filtration Rate by CKD-EPI. IDSA: Infectious Disease Society American.

the higher risk of amputation in their population is more attributed to the metabolic status or immunosuppression of the patients [22].

**Table 2.** **Microbiological profile of the isolated bacterium according to MDRB status.**

| | Total n (%) | MDRB (+) n (%) | MDRB (-) n (%) | p-value |
|---|---|---|---|---|
| **General** | 236 (100) | 194 (82) | 42 (18) | |
| **Gram** | | | | |
| Gram-negative | 143 (61) | 121 (62) | 22 (52) | 0.230 |
| Gram-positive | 93 (39) | 73 (38) | 20 (48) | |
| **Bacterium** | | | | |
| *Enterobacteriaceae* | 121 (51.3) | 102 (52.6) | 19 (45.2) | **0.010** |
| *Pseudomonas sp.* | 12 (5.1) | 10 (5.2) | 2 (4.8) | |
| *Acinetobacter sp.* | 10 (4.2) | 9 (4.6) | 1 (2.4) | |
| *Staphylococcus sp.* | 67 (28.4) | 56 (28.9) | 11 (26.2) | |
| *Enterococcus sp.* | 21 (8.9) | 16 (8.3) | 5 (11.9) | |
| *Streptococcus sp.* | 5 (2.1) | 1 (0.5) | 4 (9.5) | |

In our study, we observed an MDRB prevalence of 82% [9], a figure significantly higher compared to other countries such as France, India, China, and Turkey, where it ranged from 13.8 to 72.5% [22]. One notable outlier was Ethiopia, where a staggering 93% prevalence was reported [23]. This high rate of MDRB in our research might be attributable to several factors. We did not exclude patients who had received prior antibiotic treatment, a practice known to foster the emergence of resistant bacterial strains [21]. In addition, 20% of our patients had a history of hospitalization due to DFI, potentially predisposing them to selective colonization by resistant pathogens [24]. Furthermore, it is essential to bear in mind that the definition of MDRB can vary across different studies, leading to discrepancies in reported prevalence rates based on the specific criteria employed.

The scientific literature suggests a difference in the prevalence of gram-positive or negative germs depending on the economic level. It is described that the prevalence of gram-negative bacteria is higher in developing countries, as in the studies by Gadepalli et al. [1] and Datta et al. [25]. Similarly, in our study, gram-negative bacteria were the most frequent, finding an 85% bone involvement and an 80% moderate-severe infection, which is related to infection by gram-negative or mixed bacteria. Also, the *Enterobacteriaceae* family presented the highest proportion of MDRB (52.6%), followed by the *Staphylococcus sp.* genus (28.9%), unlike the study by McDonald et al. [26], where *Staphylococcus sp.* was the most common MDRB. The similarities between our results and the studies from India mentioned above [1,25], could be

**Table 3.** **Clinical outcomes according to MDRB status.**

| | Total | MDRB (+) n (%) | MDRB (-) n (%) | p-value |
|---|---|---|---|---|
| **Composite primary outcome [a] (*n* = 192)** | 38 /192 (18.9) | 36/155 (23.2) | 2/37 (5.4) | **0.010** |
| **Secondary outcomes** | | | | |
| Mortality (n = 192) | 4/192 (2.1) | 3/155 (1.9) | 1/37 (2.7) | 0.770 |
| Major amputation (n = 188) | 34/188 (17.7) | 33/152 (21.7) | 1/36 (2.8) | **0.008** |
| Minor amputation [b] (n = 154) | 82/154 (53.2) | 65/119 (54.6) | 17/35 (48.6) | 0.530 |
| Inpatient care [b] (n = 154) | 90/154 (67.0) | 66/119 (55.5) | 24/35 (68.6) | 0.170 |
| Prolonged hospitalization [c] (n = 90) | 66/90 (80.0) | 53/68 (80.3) | 13/24 (54.2) | **0.013** |

[a] Primary composite outcome: Mortality and/or major amputation. [b] Not include fatalities or major amputations. [c] longer than 14 days.

**Table 4. Association between primary composite outcome and population characteristics.**

| Variable | Primary composite outcome [a] (+) n (%) | Primary composite outcome [a] (-) n (%) | *p*-value |
|---|---|---|---|
| **General** | 38 (19.8) | 154 (80.2) | |
| **Demographic** | | | |
| Male | 25 (18) | 117 (82) | 0.200 |
| Age $\geq$ 60 years | 31 (32) | 65 (68) | **0.001** |
| DM duration (years) $\geq$ 10 | 32 (24) | 101 (76) | **0.046** |
| **Diabetic foot ulcer** | | | |
| Wagner classification | | | |
| 1 | 0 (0) | 5 (100) | **0.004** |
| 2 | 1 (6) | 17 (94) | |
| 3 | 8 (11) | 64 (89) | |
| 4 | 29 (30) | 68 (70) | |
| IDSA infection grade | | | |
| Mild | 2 (9) | 21 (91) | **<0.001** |
| Moderate | 17 (14) | 105 (86) | |
| Severe | 19 (40) | 28 (60) | |
| Peripheral arterial disease | 29 (41) | 42 (59) | **0.001** |
| Re-ulceration | 15 (2) | 39 (72) | 0.080 |
| Area $\geq$ 10 cm$^2$ | 32 (2) | 98 (75) | **0.010** |
| Peripheral neuropathy | 38 (20) | 148 (80) | 0.210 |
| San Elian score $\geq$ 20 | 22 (31) | 48 (69) | **0.002** |
| **Comorbidities** | | | |
| Previous major amputation due to diabetic foot | 9 (65) | 5 (35) | **<0.001** |
| Any previous minor amputation due to diabetic foot | 11 (17) | 55 (83) | 0.410 |
| Any previous diabetic foot hospitalization | 10 (28) | 26 (72) | 0.180 |
| Antibiotic therapy within the last year | 19 (28) | 49 (72) | **0.030** |
| Diabetic foot infection within the last year | 15 (24) | 47 (76) | 0.290 |
| Chronic kidney disease <60 mil/min | 7 (23) | 24 (77) | 0.840 |
| **Laboratory Findings** | | | |
| Hemoglobin < 10 g/dl | 21 (32) | 44 (68) | **0.006** |
| Albumin (g/dL) <3.5 g/dL | 29 (33) | 60 (67) | **0.007** |
| HbA1c $\geq$ 7% | 20 (16) | 108 (85) | **0.003** |
| **Microbiological** | | | |
| Polymicrobial | 13 (35) | 24 (65) | **0.009** |
| Multidrug resistant organism | | | |
| *Staphylococcus sp.*[b] | 12 (23) | 40 (77) | 0.150 |
| *Enterococcus sp.*[c] | 5 (31) | 11 (69) | 0.200 |
| *Enterobacteriaceae*[d] | 22 (24) | 69 (76) | 0.260 |
| *Pseudomonas* sp.[e] | 2 (20) | 8 (80) | 0.490 |
| *Acinetobacter sp.*[f] | 2 (25) | 6 (75) | 0.570 |

[a] Primary outcome: Mortality and/or major amputation. Total of each bacterium [b] n = 67 [c] n = 20 [d] n = 108 [e] n = 12 [f] n = 9.

related to the similar degree of hygiene, health education, footwear use, and geographical area related to warm climates. It could also be related to the low budget that the health system invests in creating programs aimed at improving the management and treatment of patients with DFI.

**Table 5. Association between MDRB and clinical outcomes: Crude and adjusted models.**

| | Model | Crude | p-value | Model 1* | Adjusted | p-value | Model 2** | Adjusted | p-value | Model 3*** | Adjusted | p-value |
|---|---|---|---|---|---|---|---|---|---|---|---|---|
| | RR | 95% CI | | RR | 95% CI | | RR | 95% CI | | RR | 95% CI | |
| **Composite primary outcome [a]** | | | | | | | | | | | | |
| Not MDRB | 1.00 | | | 1.00 | | | 1.00 | | | 1.00 | | |
| MDRB | 4.29 | 1.08–17.1 | 0.039 | 3.40 | 0.86–13.4 | 0.081 | 3.28 | 0.99–10.8 | 0.051 | 3.29 | 0.77–13.9 | 0.105 |
| **Secondary outcomes** | | | | | | | | | | | | |
| **Mortality** | | | | | | | | | | | | |
| Not MDRB | 1.00 | | | 1.00 | | | 1.00 | | | 1.00 | | |
| MDRB | 0.71 | 0.08–6.72 | 0.770 | 0.27 | 0.04–1.96 | 0.194 | 0.53 | 0.05–5.91 | 0.611 | 0.27 | 0.05–1.66 | 0.158 |
| **Major amputation** | | | | | | | | | | | | |
| Not MDRB | 1.00 | | | 1.00 | | | 1.00 | | | 1.00 | | |
| MDRB | 7.81 | 1.1–55.5 | 0.040 | 6.45 | 0.91–45.4 | 0.061 | 5.41 | 0.88–33.2 | 0.068 | 6.55 | 0.89–47.9 | 0.064 |
| **Minor amputation [b]** | | | | | | | | | | | | |
| Not MDRB | 1.00 | | | 1.00 | | | 1.00 | | | 1.00 | | |
| MDRB | 1.12 | 0.76–1.64 | 0.544 | 1.15 | 0.78–1.69 | 0.474 | 0.98 | 0.68–1.40 | 0.908 | 1.20 | 0.81–1.77 | 0.350 |
| **Inpatient care [b]** | | | | | | | | | | | | |
| Not MDRB | 1.00 | | | 1.00 | | | 1.00 | | | 1.00 | | |
| MDRB | 0.81 | 0.61–1.07 | 0.133 | 0.84 | 0.64–1.11 | 0.222 | 0.77 | 0.59–1.01 | 0.059 | 0.81 | 0.59–1.10 | 0.182 |
| **Longer hospitalization [c]** | | | | | | | | | | | | |
| Not MDRB | 1.00 | | | 1.00 | | | 1.00 | | | 1.00 | | |
| MDRB | 1.49 | 1.06–2.10 | 0.020 | 1.45 | 1.05–2.02 | 0.026 | 1.46 | 1.06–2.02 | 0.021 | 1.51 | 1.06–2.14 | 0.021 |

[a]Primary composite outcome: Mortality and/or major amputation. [b]Does not include deaths or major amputation ($n = 154$). [c] Greater than 14 days ($n = 90$).

RR: Relative risk. CI: Confidence interval. *Model 1: Adjusted for age and gender. **Model 2: Adjusted for bone compromise, severe infection, and peripheral arterial disease. *** Model 3: Adjusted for diabetes mellitus duration and HbA1C.

Prolonged hospitalization was the only adverse clinical outcome associated with MDRB, a finding consistent with previous studies [8,24], which could suggest nosocomial acquisition through cross-transmission via caregivers' hands [27,28]. The search for hospital care in these patients, as well as the extended stay, may be influenced by factors such as the complexity of the procedures, the longer duration of ulcers requiring prolonged antibiotic treatment, timely follow-up, and slow recovery in patients with MDRB [8]. Furthermore, it is noteworthy that 50.5% of patients had a Wagner classification of 4 and 88% had a moderate-severe infection according to IDSA guideline, so it was expected that a longer hospitalization would be due to the severity of their clinical condition [29].

In our study, several factors were associated with the primary outcome, including polymicrobial culture, high scores in prediction clinical rules, infection severity, older age, longer duration of diabetes, PAD history, larger wound area, low hemoglobin and albumin levels, and high glycated hemoglobin. Many of these factors are well-recognized indicators of a poor prognosis and of causing confusion or modification of the relationship between MDRB and death or major amputation.

Polymicrobial culture has been associated with the severity of DFI. The symbiosis between two bacteria can exacerbate tissue damage compared to when they are individually present. The severity scales described—Wagner, San Elian, and IDSA—are directly proportional to a worse outcome. Mild infections are described as being monomicrobial and generally gram-positive. However, as the severity of the scale increases, it is usually polymicrobial and with the presence of gram-negatives [30]. Therefore, the scales indicate the need for broad-spectrum

treatment when encountering severe infections. Vascular insufficiency due to PAD leads to tissue hypoxia and a reduction in antibiotic concentrations at the infection site. Thus, longer diabetes mellitus duration or older age may result in increased arterial atherosclerosis and greater PAD, thereby increasing the risk for adverse clinical outcomes [31]. Similarly, anemia contributes to poor wound healing by reducing oxygen delivery to damaged tissues. Lower hemoglobin and albumin would indicate a worse condition to face the infection, likewise a state of prolonged infection exacerbates catabolism that worsens the condition of anemia and chronic malnutrition [32].

Regarding the degree of glycemic control, the population was not similarly distributed in both groups, making it challenging to analyze its relationship with adverse outcomes accurately. However, most patients with DFI in our study had suboptimal glycemic control (HbA1c greater than 7% in 77.6% of patients with MDRB and 93.9% of non-MDRB patients), with an average HbA1c above 9%. The high prevalence of anemia in our study further complicates the interpretation and influence of this variable [13].

## Importance in public health

Our study revealed a significant high rate of MDRB, much higher than most reports worldwide. This high rate could distort the impact on outcomes. Given this reality, it is critical that all health facilities establish their bacteriological profile and susceptibility for DFI. This would enable early and effective empirical antibiotic use tailored to the specific profile, preventing delays due using of less effective antibiotics that could contribute to infection progression.

Many MDRB patients, due to intravenous treatments, require continued hospitalization, thereby increasing the intensive use of healthcare personnel. Extended stays heighten the risk of nosocomial infections, escalate direct healthcare costs, and reduce admission availability for other patients. The finding of an association with MDRB suggests an extreme need for rigorous biosecurity measures to prevent cross-contamination and fomites from healthcare personnel and other patients.

## Limitations and strengths

The high prevalence of patients with MDRB could have limited their influence on outcomes in our cross-sectional analysis. We needed more non-MDRB patients to achieve a close-to-1 exposed/non-exposed ratio and improve estimation precision. Another limitation is the lack of cultures for anaerobic bacteria, which could have altered the distribution of germs in each group (MDRB and non-MDRB) when assessing the association. Additionally, our study's findings cannot be generalized to other populations due to the presence of a Diabetic Foot program, which includes a multidisciplinary team that standardizes treatment, performing surgical debridement within the first 48 hours, contributing to the low rate of adverse DFI clinical outcomes observed. Finally, a retrospective cohort study based on registries could limit the accuracy of the measurements. However, the Diabetic Foot program follows international guidelines and has standardized questionnaires and completion forms.

Key strengths include the use of antibiotic therapy as per the IDSA guidelines and the procedures of the International Working Group of Diabetic Foot for managing PAD, offloading, glycemic management, and nutrition. Also noteworthy is our adherence to recommended standards for sample collection to avoid contamination and the use of a modern system for evaluating bacterial susceptibility.

## Conclusion

Our study did not find an association between MDRB and death and/or major amputation, but it did with hospitalization longer than 14 days. The high prevalence of MDRB in the total sample and the cross-sectional design may have limited the discovery of differences in the primary outcome. Further prospective studies, with larger sample sizes and equivalent studies, are needed to confirm its effect on these outcomes. The high prevalence of MDRB highlights the need for strategies to improve the timely identification of diabetic foot ulcers, as well as proper sample collection to identify pathogens and determine their susceptibility pattern to antibiotics before treatment initiation.

## Supporting information

**S1 Fig. Diagram of distribution of outcomes in the study's patients.**
(DOCX)

**S1 Table. Categories and agents used to define *Staphylococcus aureus* MDR, XDR, and PDR.**
(DOCX)

**S2 Table. Categories and agents used to define *Enterococcus sp* MDR, XDR and PDR.**
(DOCX)

**S3 Table. Categories and agents used to define *Enterobacteriaceae* MDR, XDR and PDR.**
(DOCX)

**S4 Table. Categories and agents used to define *Pseudomonas aeuruginosa* MDR, XDR and PDR.**
(DOCX)

**S5 Table. Categories and agents used to define *Acinetobacter spp* MDR, XDR and PDR.**
(DOCX)

**S6 Table. Scales to assess the characteristics of DFI.**
(DOCX)

**S7 Table. Current Procedimental Terminology codes for major amputations.**
(DOCX)

**S8 Table. Data collection instrument.**
(DOCX)

**S1 Dataset.**
(XLS)

## Author Contributions

**Conceptualization:** Marlon Yovera-Aldana, Paola Sifuentes-Hermenegildo.

**Data curation:** Marlon Yovera-Aldana.

**Formal analysis:** Marlon Yovera-Aldana, Paola Sifuentes-Hermenegildo, Edward Mezones-Holguin.

**Investigation:** Marlon Yovera-Aldana, Paola Sifuentes-Hermenegildo.

**Methodology:** Martha Sofia Cervera-Ocaña, Edward Mezones-Holguin.

**Project administration:** Marlon Yovera-Aldana.

**Supervision:** Marlon Yovera-Aldana.

**Validation:** Marlon Yovera-Aldana, Paola Sifuentes-Hermenegildo.

**Writing – original draft:** Marlon Yovera-Aldana, Paola Sifuentes-Hermenegildo, Martha Sofia Cervera-Ocaña, Edward Mezones-Holguin.

**Writing – review & editing:** Marlon Yovera-Aldana, Paola Sifuentes-Hermenegildo, Martha Sofia Cervera-Ocaña, Edward Mezones-Holguin.

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
