## [Decision Letter · Decision Letter 0]

8 Mar 2024

PONE-D-24-05269Association of multidrug-resistant bacteria and clinical outcomes in patients with infected diabetic foot in a Peruvian Hospital: A retrospective cohort analysisPLOS ONE

Dear Dr. Yovera-Aldana,

Thank you for submitting your manuscript to PLOS ONE. After careful consideration, we feel that it has merit but does not fully meet PLOS ONE’s publication criteria as it currently stands. Therefore, we invite you to submit a revised version of the manuscript that addresses the points raised during the review process.

We look forward to receiving your revised manuscript.

Kind regards,

Chikezie Hart Onwukwe

Academic Editor

PLOS ONE

Journal Requirements:

Reviewers' comments:

Reviewer's Responses to Questions

**Comments to the Author**

1. Is the manuscript technically sound, and do the data support the conclusions?

Reviewer #1: No

Reviewer #2: Yes

2. Has the statistical analysis been performed appropriately and rigorously? 

Reviewer #1: No

Reviewer #2: Yes

3. Have the authors made all data underlying the findings in their manuscript fully available?

Reviewer #1: No

Reviewer #2: Yes

4. Is the manuscript presented in an intelligible fashion and written in standard English?

Reviewer #1: No

Reviewer #2: Yes

5. Review Comments to the Author

Reviewer #1: Dear Authors,

Although it is not an original study, diabetic foot is one of the most devastating complications of diabetes, and multidrug-resistant bacteria is also a significant problem worldwide. This manuscript deserves some revision, as pointed out below:

# As a retrospective cohort study, the data may be of poor quality.

#The sample used for the evaluation corresponded to 32 % of the patients in the database of the DFU, and ~50% of the patients excluded had no culture.

#In the item 'Sample Size,' consider explaining in the writing what you considered 'exposed' and 'non-exposed.'

# It needed to be more precise on how you collected the data. Did you use any tools? Does the DFU use any standardized protocol to classify the severity and decide whether hospitalization or outpatient management is necessary?

#In the item 'Adverse Clinical Outcomes', the description of the secondary outcomes includes the primary composite outcome. What is the reason for joining death and major amputation as a primary outcome and all the outcomes separated as a secondary outcome? That is different from the diagram at the supporting data. Also the number that is 38 and not 37 at the primary outcome and 154 as 'other outcome.' In the supporting data, it says non-major and non-minor amputation; what's the difference? And what you consider ' hospitalization solely for medical treatment' needs to be clarified. A detailed explanation of the data collection process is essential as it allows other researchers to replicate your study and verify your findings, thereby enhancing the credibility of your work.

#In the item ' Diabetic Foot Characteristics', you must detail how you got these data. It seems you applied the San Elian scale using the recorded data. Did all patients have plethysmography reports? You also classified the degree of infection using IDSA Consensus. It is essential to describe whether the DFU follows a standardized protocol.

# In the Item ' Microbiological Analysis,' the biopsy is described as a prospective study, not a retrospective. It needs clarification.

# In the results, you included the Wagner classification; you need to report you would include this classification in the methods. And clarify why there is a table in the supporting data listing all the other scales used in the study. And it is important to remember that different scales may lead to different classifications.

# Consider commenting on the majority(~80%) of patients who had primary composite outcome were >60years old, had a longer duration of diabetes, low albumin and Wagner classification 4, besides ~50% of them had severe IDSA, Hb <10g/dl , San Elian score >=20 and previous antibiotic therapy.

# It is also necessary to review the writing for a few mistakes. I suggest not using the abbreviation IDF, as it may be confused with the International Diabetes Federation.

Reviewer #2: The manuscript was well-written, detailed and easy to understand. Data has been provided clearly, fully available and it supports the conclusion. The authors have done a good job with the manuscript-i recommend publication.

6. PLOS authors have the option to publish the peer review history of their article (what does this mean?). If published, this will include your full peer review and any attached files.

Reviewer #1: No

Reviewer #2: No

---

## [Author Response · Author response to Decision Letter 0]

18 Apr 2024

We appreciate the comments made by the reviewer. Below we proceed to answer one by one.

1.# As a retrospective cohort study, the data may be of poor quality.

We agree that conducting a retrospective cohort study may introduce limitations in data quality, as noted in our limitations. However, we believe that this data serves to generate hypotheses, particularly in resource-limited settings like ours.

Now it says in limitations:

“Finally, a retrospective cohort study based on registries could limit the accuracy of the measurements. However, the Diabetic Foot program follows international guidelines and has standardized questionnaires and completion forms.”

2.#The sample used for the evaluation corresponded to 32 % of the patients in the database of the DFU, and ~50% of the patients excluded had no culture.

Patients who did not present a culture had non-infected lesions. Likewise, many lesions only present mild cellulitis-type soft tissue infection or mild infected ulcers with no history of antibiotic treatment, which were not requested either.. The clinical guideline used, based on the 2010 IDSA-IWGDF Guideline, which remains current as of 2023, supports the notion that empirical treatment without culture may be justified (1,2)

1. Lipsky BA, Berendt AR, Cornia PB, Pile JC, Peters EJ, Armstrong DG, Deery HG, Embil JM, Joseph WS, Karchmer AW, Pinzur MS, Senneville E; Infectious Diseases Society of America. 2012 Infectious Diseases Society of America clinical practice guideline for the diagnosis and treatment of diabetic foot infections. Clin Infect Dis. 2012 Jun;54(12):e132-73. doi: 10.1093/cid/cis346. PMID: 22619242.

2. Senneville É, Albalawi Z, van Asten SA, Abbas ZG, Allison G, Aragón-Sánchez J, Embil JM, Lavery LA, Alhasan M, Oz O, Uçkay I, Urbančič-Rovan V, Xu ZR, Peters EJG. IWGDF/IDSA guidelines on the diagnosis and treatment of diabetes-related foot infections (IWGDF/IDSA 2023). Diabetes Metab Res Rev. 2024 Mar;40(3):e3687. doi: 10.1002/dmrr.3687. Epub 2023 Oct 1. PMID: 37779323.

3. #In the item 'Sample Size,' consider explaining in the writing what you considered 'exposed' and 'non-exposed.'

We have replaced "exposed" with "subjects with MDR" and "unexposed" with "subjects without MDR".

It said:

After applying the inclusion and exclusion criteria, we obtained a sample size of 192 patients. The power of the sample was 88%, considering a proportion of 82% of the primary outcome in exposed subjects, 78% in non-exposed subjects, a confidence level of 95%, and a 4:1 ratio between exposed and non-exposed subjects.

Now, it says:

After applying the inclusion and exclusion criteria, we obtained a sample size of 192 patients. The sample's statistical power was 88%, calculated based on an 82% proportion of the primary outcome among subjects with MDR and 78% in subjects without MDR, at a confidence level of 95%, with a 4:1 ratio between MDR and non-MDR subjects.

4. # It needed to be more precise on how you collected the data. Did you use any tools? Does the DFU use any standardized protocol to classify the severity and decide whether hospitalization or outpatient management is necessary?

We have specified that the diabetic foot unit service follows the IWGDF guidelines. Additionally, we have provided greater detail of the processes carried out by the DFU team. Also, the DFU maintains standardized formats for evaluating and monitoring patients until they achieve epithelialization, undergo amputation, or experience death.

Now, It says:

At the María Auxiliadora hospital, patients with acute infection, whether admitted through the clinic or through the emergency department, are preferentially managed by a member of the DFU team. The decision for hospitalization or outpatient care depends on the severity of the infection, the patient's age, and environmental factors such as family support.

Upon admission, all patients undergo a wound evaluation according to standardized formats. An arterial Doppler ultrasound of the affected limb is performed to assess arterial waveform characteristics related to the injury. As per protocol, the infected lesion is debrided witin the first 48 hours after admission; However, swab cultures are not performed. Only transportation and cultivation methods for aerobic strains are available. The culture is taken by biopsy of the debrided wound bed. Transportation is carried out in sterile warehouses. The antibiotic is started according to the IDSA scale for empirical treatment. Antibiotics are adjusted based on culture results obtained after 72 hours.. During hospitalization, the patient receives wound care three to four times weekly by the DFU team, with bedside surgical debridement and advanced wound healing using silver hydrogels or other similar modalities.

5. #In the item 'Adverse Clinical Outcomes', the description of the secondary outcomes includes the primary composite outcome. What is the reason for joining death and major amputation as a primary outcome and all the outcomes separated as a secondary outcome? That is different from the diagram at the supporting data. 

In 2020, Armstrong reported that major leg amputation was closely associated with poor 5-year survival, comparable to survival rates seen in cancers (1). Since then, some authors have evaluated this composite primary outcome (2,3). In our point of view, this primary outcome serves its purpose effectively due to its high clinical significance (4).

1. Armstrong DG, Swerdlow MA, Armstrong AA, Conte MS, Padula WV, Bus SA. Five year mortality and direct costs of care for people with diabetic foot complications are comparable to cancer. J Foot Ankle Res. 2020 Mar 24;13(1):16. doi: 10.1186/s13047-020-00383-2. PMID: 32209136; PMCID: PMC7092527.

2. Brennan MB, Powell WR, Kaiksow F, Kramer J, Liu Y, Kind AJH, Bartels CM. Association of Race, Ethnicity, and Rurality With Major Leg Amputation or Death Among Medicare Beneficiaries Hospitalized With Diabetic Foot Ulcers. JAMA Netw Open. 2022 Apr 1;5(4):e228399. doi: 10.1001/jamanetworkopen.2022.8399. PMID: 35446395; PMCID: PMC9024392.

3. Chamberlain RC, Fleetwood K, Wild SH, Colhoun HM, Lindsay RS, Petrie JR, McCrimmon RJ, Gibb F, Philip S, Sattar N, Kennon B, Leese GP. Foot Ulcer and Risk of Lower Limb Amputation or Death in People With Diabetes: A National Population-Based Retrospective Cohort Study. Diabetes Care. 2022 Jan 1;45(1):83-91. doi: 10.2337/dc21-1596. PMID: 34782354.

4. Baracaldo-Santamaría D, Feliciano-Alfonso JE, Ramirez-Grueso R, Rojas-Rodríguez LC, Dominguez-Dominguez CA, Calderon-Ospina CA. Making Sense of Composite Endpoints in Clinical Research. J Clin Med. 2023 Jun 29;12(13):4371. doi: 10.3390/jcm12134371. PMID: 37445406; PMCID: PMC10342974.

Also the number that is 38 and not 37 at the primary outcome and 154 as 'other outcome.' 

The diagram shows the classification according to the exhibition that coincides with the presented text. Number 37 are the people who did not present MDR (not exposed). Number 38 are the people with the primary result, 36 from the exposed group and 2 from the non-exposed group. We make greater detail in the diagram.

In the supporting data, it says non-major and non-minor amputation; what's the difference? And what you consider ' hospitalization solely for medical treatment' needs to be clarified. A detailed explanation of the data collection process is essential as it allows other researchers to replicate your study and verify your findings, thereby enhancing the credibility of your work.

We added more details about the evaluated clinical outcomes.

It said:

“The primary composite outcome consisted of death due to IDF complications and/or major amputation (above the ankle). Secondary outcomes were death related to the diabetic foot, major amputation, minor amputation (below the ankle), hospitalization solely for medical treatment, and prolonged stay (more than 14 days).”

Now it says:

The primary outcome was a composite of major leg amputation or death during hospitalization or within the first 30 days after discharge. Any surgery above the ankle was considered a major amputation. In Table S7, we present the codes for procedures classified as major amputation according to the Current Procedural Terminology (CPT). As a secondary outcome, we assessed the individual components of the primary outcome. We also considered minor leg amputations, which included phalangeal disarticulation, beam amputation, transmetatarsal amputation, Lisfranc and Syme amputation. Another secondary outcome was the indication for inpatient care for intravenous antibiotic treatment and prolonged hospitalization (hospital stays longer than 14 days).

As Supplementary Table 7, we added CPT codes that include the major leg amputation procedure:

Major amputation CPT Code Description of Procedure

Below knee amputation 27880 Amputation leg through tibia and fibula

 27881 Amputation leg through the tibia and fibula with immediate fitting technique including application of first cast

 27882 Amputation leg through the tibia and fibula, open circulatory (guillotine)

 27886 Amputation leg through the tibia and fibula, re-amputation

Above knee amputation 27295 Disarticulation of hip

 27590 Amputation, thigh, through femur, any level

 27591 Amputation, thigh, through the femur, any level, an immediate fitting technique including the first cast

 27592 Amputation, thigh, through femur, any level, open, circular (guillotine)

 27596 Amputation, thigh, through femur, any level, re-amputation

 27598 Disarticulation at the knee

6. #In the item ' Diabetic Foot Characteristics', you must detail how you got these data. It seems you applied the San Elian scale using the recorded data. Did all patients have plethysmography reports? You also classified the degree of infection using IDSA Consensus. It is essential to describe whether the DFU follows a standardized protocol.

We provide a better description in the methods section of the variables included in the tables. The change can be seen in item 8.

We have specified that the diabetic foot unit service follows the IWGDF guidelines. We have provided greater detail of the processes carried out by the DFU team. Please refer to Item 4 for these updates

7. # In the Item ' Microbiological Analysis,' the biopsy is described as a prospective study, not a retrospective. It needs clarification.

We made better phrasing about this.

It said:

The DFU staff obtained culture samples via tissue biopsy following the wound debridement using the standard procedure.

Now, It says:

Culture results were obtained from clinical records, encompassing all identified strains. The endocrinology service staff performed culture samples via tissue biopsy following wound debridement using the standard procedures.

8. # In the results, you included the Wagner classification; you need to report you would include this classification in the methods. 

We provide a better description in the methods section of the variables included in the tables. 

It said:

We described the characteristics of the lesion and the patient upon admission to the DFU using the San Elian scale. This tool includes 10 variables: description of depth, infection, ischemia, neuropathy, location, topography, number of zones, edema, and degree of inflammation[15]. The largest and smallest diameters of the ulcer were recorded, and the area was calculated using the ellipse formula. The degree of infection was based on the IDSA consensus[10]. Peripheral arterial disease classification was obtained from arterial plethysmography reports, with positive results if biphasic, monophasic, or absence of wave patterns were present. Peripheral neuropathy was diagnosed if there was a loss of protective sensation evaluated with a monofilament. The glomerular filtration was calculated using the CKD-EPI formula[16]. Previous exposure to antibiotics in the last year, as well as previous amputation or hospitalization for diabetic foot, was documented. Furthermore, we recorded the levels of hemoglobin, albumin, leukocytes, and creatinine at the time of the acute episode. We accepted HbA1c values up to one month old.

Now, It says:

We described the demographics and clinical characteristics of lesions and patients upon admission to the DFU. In the demographic variables, we described gender (male/female), age (≥ 60/< 60 years); and duration of diabetes (≥ 10 years/ < 10 years). Regarding foot characteristics, we described the Wagner classification (stage 1, 2, 3, 4); infection severity based on the Infectious Disease Of Society American (IDSA) Guideline of DFI (mild, moderate, severe) [10]; peripheral arterial disease (PAD) determined from arterial plethysmography reports (positive result if previous history or biphasic, monophasic, or absence of wave pattern/negative if triphasic pattern); type of ulcer (new/ reulceration); ulcer area calculated using the ellipse formula (≥ 10 cm2 /< 10 cm2); peripheral neuropathy (Yes if previous history or recorded as loss of protective sensation evaluated with a monofilament). Additionally, we described the San Elian score, which includes description of depth, infection, ischemia, neuropathy, location, topography, number of zones, edema, and degree of injury inflammation (moderate or < 20 / severe or ≥20) [15] In Supplementary Table S6 For informational purposes, we describe the components and categories of the scales used and others that were not available in S7 Table.

We recorded the following clinical history details: previous major amputation or surgery above the ankle (yes/not); any history of minor amputation or surgery below the ankle due to diabetic foot issues (yes /not); any previous hospitalization for diabetic foot-releated reasons (yes/not); previous antibiotic therapy within the last year (yes/not, ); and previous DFI within the last year (yes/not). Also, we recorded previous diabetic retinopathy (yes/no ); previous chronic kidney disease by CKD-EPI formula (≥60 ml/min/24 h /< 60 ml/min/24 h) [16], hospitalization duration and total follow-up period in days.

In the laboratory results, we described hemoglobin (< 10/ ≥ 10 g/dl); albumin (< 3.5/ ≥ 3.5 g/dl); and glycosylated hemoglobin (< 7/ ≥ 7%). HbA1c values up to one month old were accepted. Additionally, we recorded the number of bacterial strains (monomicrobial/ polymicrobial). (S6 Table).

And clarify why there is a table in the supporting data listing all the other scales used in the study. And it is important to remember that different scales may lead to different classifications.

In table S6, we place the operational definitions of the other scales only for informational purposes, allowing readers to observe the differences in criteria and categories compared to those used in our study. For our analysis, we only extracted the classification of Wagner and San Elian as the others were not available. We added an explanation in the footer of the table detailing this intention.

9. # Consider commenting on the majority(~80%) of patients who had primary composite outcome were >60years old, had a longer duration of diabetes, low albumin and Wagner classification 4, besides ~50% of them had severe IDSA, Hb <10g/dl , San Elian score >=20 and previous antibiotic therapy.

We elaborated further on the discussion.

It said:

Other factors that modify the relationship between the presence of MDRB and adverse outcomes are polymicrobial culture, vascular insufficiency, anemia, and hyperglycemia. Polymicrobial culture has been associated with the severity of DFI. The symbiosis between two bacteria can potentiate the damage when they are individually present[30]. It is also suggested that vascular insufficiency leads to tissue hypoxia and also causes a decrease in antibiotic concentrations at the site[31]. Similarly, anemia is associated with a decrease in the supply of oxygen to damaged tissues, leading to poor wound healing[32]. Regarding the degree of glycemic control, the population was not similarly distributed in both groups, which does not allow an accurate analysis of the relationship of this variable with adverse outcomes. However, most patients with DFI

---

## [Decision Letter · Decision Letter 1]

1 May 2024

PONE-D-24-05269R1Association of multidrug-resistant bacteria and clinical outcomes in patients with infected diabetic foot in a Peruvian Hospital: A retrospective cohort analysisPLOS ONE

Dear Dr. Yovera-Aldana,

Thank you for submitting your manuscript to PLOS ONE. After careful consideration, we feel that it has merit but does not fully meet PLOS ONE’s publication criteria as it currently stands. Therefore, we invite you to submit a revised version of the manuscript that addresses the points raised during the review process.

We look forward to receiving your revised manuscript.

Kind regards,

Chikezie Hart Onwukwe

Academic Editor

PLOS ONE

Journal Requirements:

Reviewers' comments:

Reviewer's Responses to Questions

**Comments to the Author**

1. If the authors have adequately addressed your comments raised in a previous round of review and you feel that this manuscript is now acceptable for publication, you may indicate that here to bypass the “Comments to the Author” section, enter your conflict of interest statement in the “Confidential to Editor” section, and submit your "Accept" recommendation.

Reviewer #1: All comments have been addressed

2. Is the manuscript technically sound, and do the data support the conclusions?

Reviewer #1: Yes

3. Has the statistical analysis been performed appropriately and rigorously? 

Reviewer #1: Yes

4. Have the authors made all data underlying the findings in their manuscript fully available?

Reviewer #1: Yes

5. Is the manuscript presented in an intelligible fashion and written in standard English?

Reviewer #1: No

6. Review Comments to the Author

Reviewer #1: Dear Authors,

Diabetic foot infection is still a significant complication of diabetes, and the manuscript addresses multidrug-resistant bacteria, a crescent and worrying problem. Despite the limitations of a retrospective study and other limitations pointed out by the authors, it reinforces that a planned and prompt treatment strategy may provide better outcomes even with multidrug-resistant bacteria and the need for prospective and more extensive studies. The authors satisfactorily addressed the reviewers' comments, and the corrections facilitated the understanding. However, some things could be improved in the writing of the whole manuscript, mainly related to grammar, punctuation, and misspellings.

Line 24: idiabetic

Line 118: Lisfranc and Syme amputation- Lisfranc, and Syme amputation.

Line 124: (≥ 60/< 60 years); (≥ 60/< 60 years),

Line 129: (new/ reulceration) (new/ re-ulceration)

Line 138- 142: (yes/not) (yes/no)

Line 140: releated related

Line 148: polymicrobial)..

Line 161: admitted through the clinic or through the emergency -admitted through the clinic or the emergency

Line 173: by - from

Line 189: percentages, percentages

Line 301: itis

Line 313: of...with between...and

Line 316: to create in creating

Line 338: tissuedamage

compare compared

Line 341: gram positive gram-positive

Line 342 gram negative gram-negative

Line 346: arterialatherosclerosis

Line 361: itis

Line 369: availabilityfor

Line 387:theuse

Line 394: find association find an association

Line 397: main primary

7. PLOS authors have the option to publish the peer review history of their article (what does this mean?). If published, this will include your full peer review and any attached files.

Reviewer #1: No

---

## [Author Response · Author response to Decision Letter 1]

2 May 2024

Dear reviewer, we appreciate the time in reviewing the work. 

Diabetic foot infection is still a significant complication of diabetes, and the manuscript addresses multidrug-resistant bacteria, a crescent and worrying problem. Despite the limitations of a retrospective study and other limitations pointed out by the authors, it reinforces that a planned and prompt treatment strategy may provide better outcomes even with multidrug-resistant bacteria and the need for prospective and more extensive studies. The authors satisfactorily addressed the reviewers' comments, and the corrections facilitated the understanding. However, some things could be improved in the writing of the whole manuscript, mainly related to grammar, punctuation, and misspellings.

Line 24: idiabetic

Line 118: Lisfranc and Syme amputation- Lisfranc, and Syme amputation.

Line 124: (≥ 60/< 60 years); (≥ 60/< 60 years),

Line 129: (new/ reulceration) (new/ re-ulceration)

Line 138- 142: (yes/not) (yes/no)

Line 140: releated related

Line 148: polymicrobial)..

Line 161: admitted through the clinic or through the emergency -admitted through the clinic or the emergency

Line 173: by - from

Line 189: percentages, percentages

Line 301: itis

Line 313: of...with between...and

Line 316: to create in creating

Line 338: tissuedamage

compare compared

Line 341: gram positive gram-positive

Line 342 gram negative gram-negative

Line 346: arterialatherosclerosis

Line 361: itis

Line 369: availabilityfor

Line 387:theuse

Line 394: find association find an association

Line 397: main primary

Answer: We correct each of the indicated lines and a complete new revision of the text was made.

---

## [Decision Letter · Decision Letter 2]

13 May 2024

Association of multidrug-resistant bacteria and clinical outcomes in patients with infected diabetic foot in a Peruvian Hospital: A retrospective cohort analysis

PONE-D-24-05269R2

Dear Dr. Marlon Yovera-Aldana,

We’re pleased to inform you that your manuscript has been judged scientifically suitable for publication and will be formally accepted for publication once it meets all outstanding technical requirements.

Kind regards,

Chikezie Hart Onwukwe

Academic Editor

PLOS ONE

Additional Editor Comments (optional):

Reviewers' comments:

Reviewer's Responses to Questions

**Comments to the Author**

1. If the authors have adequately addressed your comments raised in a previous round of review and you feel that this manuscript is now acceptable for publication, you may indicate that here to bypass the “Comments to the Author” section, enter your conflict of interest statement in the “Confidential to Editor” section, and submit your "Accept" recommendation.

Reviewer #1: All comments have been addressed

2. Is the manuscript technically sound, and do the data support the conclusions?

Reviewer #1: Yes

3. Has the statistical analysis been performed appropriately and rigorously? 

Reviewer #1: Yes

4. Have the authors made all data underlying the findings in their manuscript fully available?

Reviewer #1: Yes

5. Is the manuscript presented in an intelligible fashion and written in standard English?

Reviewer #1: Yes

6. Review Comments to the Author

Reviewer #1: Dear Authors,

I am glad to review this manuscript on multidrug-resistant bacteria diabetic foot infection, which is a significant complication of diabetes. Despite the limitations of a retrospective study and other limitations pointed out by the authors, it reinforces that a planned and prompt treatment strategy may provide better outcomes. The authors have satisfactorily made the corrections.

7. PLOS authors have the option to publish the peer review history of their article (what does this mean?). If published, this will include your full peer review and any attached files.

Reviewer #1: No

---

## [Editor Report · Acceptance letter]

27 May 2024

PONE-D-24-05269R2 

PLOS ONE

Dear Dr. Yovera-Aldana, 

I'm pleased to inform you that your manuscript has been deemed suitable for publication in PLOS ONE. Congratulations! Your manuscript is now being handed over to our production team.

Kind regards, 

on behalf of

Dr. Chikezie Hart Onwukwe 

Academic Editor

PLOS ONE